# Interaction Screening: Efficient and Sample-Optimal Learning of Ising Models

**Marc Vuffray**[1], **Sidhant Misra**[2], **Andrey Y. Lokhov**[1,3], and **Michael Chertkov**[1,3,4]

[1]Theoretical Division T-4, Los Alamos National Laboratory, Los Alamos, NM 87545, USA
[2]Theoretical Division T-5, Los Alamos National Laboratory, Los Alamos, NM 87545, USA
[3]Center for Nonlinear Studies, Los Alamos National Laboratory, Los Alamos, NM 87545, USA
[4]Skolkovo Institute of Science and Technology, 143026 Moscow, Russia

{vuffray, sidhant, lokhov, chertkov}@lanl.gov

## Abstract

We consider the problem of learning the underlying graph of an unknown Ising model on $p$ spins from a collection of i.i.d. samples generated from the model. We suggest a new estimator that is computationally efficient and requires a number of samples that is near-optimal with respect to previously established information-theoretic lower-bound. Our statistical estimator has a physical interpretation in terms of "interaction screening". The estimator is consistent and is efficiently implemented using convex optimization. We prove that with appropriate regularization, the estimator recovers the underlying graph using a number of samples that is logarithmic in the system size $p$ and exponential in the maximum coupling-intensity and maximum node-degree.

## 1 Introduction

A Graphical Model (GM) describes a probability distribution over a set of random variables which factorizes over the edges of a graph. It is of interest to recover the structure of GMs from random samples. The graphical structure contains valuable information on the dependencies between the random variables. In fact, the neighborhood of a random variable is the minimal set that provides us maximum information about this variable. Unsurprisingly, GM reconstruction plays an important role in various fields such as the study of gene expression [1], protein interactions [2], neuroscience [3], image processing [4], sociology [5] and even grid science [6, 7].

The origin of the GM reconstruction problem is traced back to the seminal 1968 paper by Chow and Liu [8], where the problem was posed and resolved for the special case of tree-structured GMs. In this special tree case the maximum likelihood estimator is tractable and is tantamount to finding a maximum weighted spanning-tree. However, it is also known that in the case of general graphs with cycles, maximum likelihood estimators are intractable as they require computation of the partition function of the underlying GM, with notable exceptions of the Gaussian GM, see for instance [9], and some other special cases, like planar Ising models without magnetic field [10].

A lot of efforts in this field has focused on learning Ising models, which are the most general GMs over binary variables with pairwise interaction/factorization. Early attempts to learn the Ising model structure efficiently were heuristic, based on various mean-field approximations, e.g. utilizing empirical correlation matrices [11, 12, 13, 14]. These methods were satisfactory in cases when correlations decrease with graph distance. However it was also noticed that the mean-field methods perform poorly for the Ising models with long-range correlations. This observation is not surprising

in light of recent results stating that learning the structure of Ising models using only their correlation matrix is, in general, computationally intractable [15, 16].

Among methods that do not rely solely on correlation matrices but take advantage of higher-order correlations that can be estimated from samples, we mention the approach based on sparsistency of the so-called regularized pseudo-likelihood estimator [17]. This estimator, like the one we propose in this paper, is from the class of M-estimators i.e. estimators that are the minimum of a sum of functions over the sampled data [22]. The regularized pseudo-likelihood estimator is regarded as a surrogate for the intractable likelihood estimator with an additive $\ell_1$-norm penalty to encourage sparsity of the reconstructed graph. The sparsistency-based estimator offers guarantees for the structure reconstruction, but the result only applies to GMs that satisfy a certain condition that is rather restrictive and hard to verify. It was also proven that the sparsity pattern of the regularized pseudo-likelihood estimator fails to reconstruct the structure of graphs with long-range correlations, even for simple test cases [18].

Principal tractability of structure reconstruction of an arbitrary Ising model from samples was proven only very recently. Bresler, Mossel and Sly in [19] suggested an algorithm which reconstructs the graph without errors in polynomial time. They showed that the algorithm requires number of samples that is logarithmic in the number of variables. Although this algorithm is of a polynomial complexity, it relies on an exhaustive neighborhood search, and the degree of the polynomial is equal to the maximal node degree.

Prior to the work reported in this manuscript the best known procedure for perfect reconstruction of an Ising model was through a greedy algorithm proposed by Bresler in [20]. Bresler's algorithm is based on the observation that the mutual information between neighboring nodes in an Ising model is lower bounded. This observation allows to reconstruct the Ising graph perfectly with only a logarithmic number of samples and in time quasi-quadratic in the number of variables. On the other hand, Bresler's algorithm suffers from two major practical limitations. First, the number of samples, hence the running time as well, scales double exponentially with respect to the largest node degree and with respect to the largest coupling intensity between pairs of variables. This scaling is rather far from the information-theoretic lower-bound reported in [21] predicting instead a single exponential dependency on the two aforementioned quantities. Second, Bresler's algorithm requires prior information on the maximum and minimum coupling intensities as well as on the maximum node degree, guarantees which, in reality, are not necessarily available.

In this paper we propose a novel estimator for the graph structure of an arbitrary Ising model which achieves perfect reconstruction in quasi-quartic time (although we believe it can be provably reduced to quasi-quadratic time) and with a number of samples logarithmic in the system size. The algorithm is near-optimal in the sense that the number of samples required to achieve perfect reconstruction, and the run time, scale exponentially with respect to the maximum node-degree and the maximum coupling intensity, thus matching parametrically the information-theoretic lower bound of [21]. Our statistical estimator has the structure of a consistent M-estimator implemented via convex optimization with an additional thresholding procedure. Moreover it allows intuitive interpretation in terms of what we coin the "interaction screening". We show that with a proper $\ell_1$-regularization our estimator reconstructs couplings of an Ising model from a number of samples that is near-optimal. In addition, our estimator does not rely on prior information on the model characteristics, such as maximum coupling intensity and maximum degree.

The rest of the paper is organized as follows. In Section 2 we give a precise definition of the structure estimation problem for the Ising models and we describe in detail our method for structure reconstruction within the family of Ising models. The main results related to the reconstruction guarantees are provided by Theorem 1 and Theorem 2. In Section 3 we explain the strategy and the sequence of steps that we use to prove our main theorems. Proofs of Theorem 1 and Theorem 2 are summarized at the end of this Section. Section 4 illustrates performance of our reconstruction algorithm via simulations. Here we show on a number of test cases that the sample complexity of the suggested method scales logarithmically with the number of variables and exponentially with the maximum coupling intensity. In Section 5 we discuss possible generalizations of the algorithm and future work.

## 2 Main Results

Consider a graph $G = (V, E)$ with $p$ vertexes where $V = \{1, \ldots, p\}$ is the vertex set and $E \subset V \times V$ is the undirected edge set. Vertexes $i \in V$ are associated with binary random variables $\sigma_i \in \{-1, +1\}$ that are called spins. Edges $(i, j) \in E$ are associated with non-zero real parameters $\theta_{ij}^* \neq 0$ that are called couplings. An Ising model is a probability distribution $\mu$ over spin configurations $\underline{\sigma} = \{\sigma_1, \ldots, \sigma_p\}$ that reads as follows:

$$\mu(\underline{\sigma}) = \frac{1}{Z} \exp \left( \sum_{(i,j) \in E} \theta_{ij}^* \sigma_i \sigma_j \right), \tag{1}$$

where $Z$ is a normalization factor called the partition function.

$$Z = \sum_{\underline{\sigma}} \exp \left( \sum_{(i,j) \in E} \theta_{ij}^* \sigma_i \sigma_j \right). \tag{2}$$

Notice that even though the main innovation of this paper – the efficient "interaction screening" estimator – can be constructed for the most general Ising models, we restrict our attention in this paper to the special case of the Ising models with zero local magnetic-field. This simplification is not necessary and is done solely to simplify (generally rather bulky) algebra. Later in the text we will thus refer to the zero magnetic field model (2) simply as the Ising model.

### 2.1 Structure-Learning of Ising Models

Suppose that $n$ sequences/samples of $p$ spins $\{\underline{\sigma}^{(k)}\}_{k=1,\ldots,n}$ are observed. Let us assume that each observed spin configuration $\underline{\sigma}^{(k)} = \{\sigma_1^{(k)}, \ldots, \sigma_p^{(k)}\}$ is i.i.d. from (1). Based on these measurements/samples we aim to construct an estimator $\widehat{E}$ of the edge set that reconstructs the structure exactly with high probability, i.e.

$$\mathbb{P}\left[\widehat{E} = E\right] = 1 - \epsilon, \tag{3}$$

where $\epsilon \in \left(0, \frac{1}{2}\right)$ is a prescribed reconstruction error.

We are interested to learn structures of Ising models in the high-dimensional regime where the number of observations/samples is of the order $n = \mathcal{O}(\ln p)$. A necessary condition on the number of samples is given in [21, Thm. 1]. This condition depends explicitly on the smallest and largest coupling intensity

$$\alpha := \min_{(i,j) \in E} |\theta_{ij}^*|, \ \beta := \max_{(i,j) \in E} |\theta_{ij}^*|, \tag{4}$$

and on the maximal node degree

$$d := \max_{i \in V} |\partial i|, \tag{5}$$

where the set of neighbors of a node $i \in V$ is denoted by $\partial i := \{j \mid (i, j) \in E\}$.

According to [21], in order to reconstruct the structure of the Ising model with minimum coupling intensity $\alpha$, maximum coupling intensity $\beta$, and maximum degree $d$, the required number of samples should be at least

$$n \geq \max \left( \frac{e^{\beta d} \ln \left( \frac{pd}{4} - 1 \right)}{4 d \alpha e^{\alpha}}, \frac{\ln p}{2\alpha \tanh \alpha} \right). \tag{6}$$

We see from Eq. (6) that the exponential dependence on the degree and the maximum coupling intensity are both unavoidable. Moreover, when the minimal coupling is small, the number of samples should scale at least as $\alpha^{-2}$.

It remains unknown if the inequality (6) is achievable. It is shown in [21, Thm. 3] that there exists a reconstruction algorithm with error probability $\epsilon \in \left(0, \frac{1}{2}\right)$ if the number of samples is greater than

$$n \geq \left( \frac{\beta d \left(3 e^{2\beta d} + 1\right)}{\sinh^2 (\alpha/4)} \right)^2 \left(16 \log p + 4 \ln (2/\epsilon)\right). \tag{7}$$

Unfortunately, the existence proof presented in [21] is based on an exhaustive search with the intractable maximum likelihood estimator and thus it does not guarantee actual existence of an algorithm with low computational complexity. Notice also that the number of samples in (7) scales as $\exp(4\beta d)$ when $d$ and $\beta$ are asymptotically large and as $\alpha^{-4}$ when $\alpha$ is asymptotically small.

## 2.2 Regularized Interaction Screening Estimator

The main contribution of this paper consists in presenting explicitly a structure-learning algorithm that is of low complexity and which is near-optimal with respect to bounds (6) and (7). Our algorithm reconstructs the structure of the Ising model exactly, as stated in Eq. (3), with an error probability $\epsilon \in \left(0, \frac{1}{2}\right)$, and with a number of samples which is at most proportional to $\exp(6\beta d)$ and $\alpha^{-2}$. (See Theorem 1 and Theorem 2 below for mathematically accurate statements.) Our algorithm consists of two steps. First, we estimate couplings in the vicinity of every node. Then, on the second step, we threshold the estimated couplings that are sufficiently small to zero. Resulting zero coupling means that the corresponding edge is not present.

Denote the set of couplings around node $u \in V$ by the vector $\underline{\theta}_u^* \in \mathbb{R}^{p-1}$. In this, slightly abusive notation, we use the convention that if a coupling is equal to zero it reads as absence of the edge, i.e. $\theta_{ui}^* = 0$ if and only if $(u, i) \notin E$. Note that if the node degree is bounded by $d$, it implies that the vector of couplings $\underline{\theta}_u^*$ is non-zero in at most $d$ entries.

Our estimator for couplings around node $u \in V$ is based on the following loss function coined the Interaction Screening Objective (ISO):

$$\mathcal{S}_n(\underline{\theta}_u) = \frac{1}{n} \sum_{k=1}^n \exp\left(-\sum_{i \in V \setminus u} \theta_{ui} \sigma_u^{(k)} \sigma_i^{(k)}\right). \tag{8}$$

The ISO is an empirical weighted-average and its gradient is the vector of weighted pair-correlations involving $\sigma_u$. At $\underline{\theta}_u = \underline{\theta}_u^*$ the exponential weight cancels exactly with the corresponding factor in the distribution (1). As a result, weighted pair-correlations involving $\sigma_u$ vanish as if $\sigma_u$ was uncorrelated with any other spins or completely "screened" from them, which explains our choice for the name of the loss function. This remarkable "screening" feature of the ISO suggests the following choice of the Regularized Interaction Screening Estimator (RISE) for the interaction vector around node $u$:

$$\widehat{\underline{\theta}}_u(\lambda) = \underset{\underline{\theta}_u \in \mathbb{R}^{p-1}}{\operatorname{argmin}} \mathcal{S}_n(\underline{\theta}_u) + \lambda \|\underline{\theta}_u\|_1, \tag{9}$$

where $\lambda > 0$ is a tunable parameter promoting sparsity through the additive $\ell_1$-penalty. Notice that the ISO is the empirical average of an exponential function of $\underline{\theta}_u$ which implies it is convex. Moreover, addition of the $\ell_1$-penalty preserves the convexity of the minimization objective in Eq. (9).

As expected, the performance of RISE does depend on the choice of the penalty parameter $\lambda$. If $\lambda$ is too small $\widehat{\underline{\theta}}_u(\lambda)$ is too sensitive to statistical fluctuations. On the other hand, if $\lambda$ is too large $\widehat{\underline{\theta}}_u(\lambda)$ has too much of a bias towards zero. In general, the optimal value of $\lambda$ is hard to guess. Luckily, the following theorem provides strong guarantees on the square error for the case when $\lambda$ is chosen to be sufficiently large.

**Theorem 1** (Square Error of RISE). *Let $\left\{\underline{\sigma}^{(k)}\right\}_{k=1,\ldots,n}$ be $n$ realizations of $p$ spins drawn i.i.d. from an Ising model with maximum degree $d$ and maximum coupling intensity $\beta$. Then for any node $u \in V$ and for any $\epsilon_1 > 0$, the square error of the Regularized Interaction Screening Estimator (9) with penalty parameter $\lambda = 4\sqrt{\frac{\ln(3p/\epsilon_1)}{n}}$ is bounded with probability at least $1 - \epsilon_1$ by*

$$\left\|\widehat{\underline{\theta}}_u(\lambda) - \underline{\theta}_u^*\right\|_2 \leq 2^8 \sqrt{d}(d+1)e^{3\beta d}\sqrt{\frac{\ln\frac{3p}{\epsilon_1}}{n}}, \tag{10}$$

*whenever $n \geq 2^{14} d^2 (d+1)^2 e^{6\beta d} \ln\frac{3p^2}{\epsilon_1}$.*

Our structure estimator (for the second step of the algorithm), Structure-RISE, takes RISE output and thresholds couplings whose absolute value is less than $\alpha/2$ to zero:

$$\widehat{E}(\lambda, \alpha) = \left\{(i, j) \in V \times V \mid \widehat{\theta}_{ij}(\lambda) + \widehat{\theta}_{ji}(\lambda) \geq \alpha\right\}. \tag{11}$$

Performance of the Structure-RISE is fully quantified by the following Theorem.

**Theorem 2** (Structure Learning of Ising Models). *Let* $\left\{\underline{\sigma}^{(k)}\right\}_{k=1,\ldots,n}$ *be $n$ realizations of $p$ spins drawn i.i.d. from an Ising model with maximum degree $d$, maximum coupling intensity $\beta$ and minimal coupling intensity $\alpha$. Then for any $\epsilon_2 > 0$, Structure-RISE with penalty parameter $\lambda = 4\sqrt{\frac{\ln(3p^2/\epsilon_2)}{n}}$ reconstructs the edge-set perfectly with probability*

$$\mathbb{P}\left(\widehat{E}\left(\lambda,\alpha\right) = E\right) \geq 1 - \epsilon_2, \tag{12}$$

*whenever* $n \geq \max\left(d/16, \alpha^{-2}\right) 2^{18} d \left(d+1\right)^2 e^{6\beta d} \ln \frac{3p^3}{\epsilon_2}$.

Proofs of Theorem 1 and Theorem 2 are given in Subsection 3.3.

Theorem 1 states that RISE recovers not only the structure but also the correct value of the couplings up to an error based on the available samples. It is possible to improve the square-error bound (10) even further by first, running Structure-RISE to recover edges, and then re-running RISE with $\lambda = 0$ for the remaining non-zero couplings.

The computational complexity of RISE is equal to the complexity of minimizing the convex ISO and, as such, it scales at most as $\mathcal{O}\left(np^3\right)$. Therefore, computational complexity of Structure-RISE scales at most as $\mathcal{O}\left(np^4\right)$ simply because one has to call RISE at every node. We believe that this running-time estimate can be proven to be quasi-quadratic when using first-order minimization-techniques, in the spirit of [23]. We have observed through numerical experiments that such techniques implement Structure-RISE with running-time $\mathcal{O}\left(np^2\right)$.

Notice that in order to implement RISE there is no need for prior knowledge on the graph parameters. This is a considerable advantage in practical applications where the maximum degree or bounds on couplings are often unknown.

## 3 Analysis

The Regularized Interaction Screening Estimator (9) is from the class of the so-called regularized M-estimators. Negahban et al. proposed in [22] a framework to analyze the square error of such estimators. As per [22], enforcing only two conditions on the loss function is sufficient to get a handle on the square error of an $\ell_1$-regularized M-estimator.

The first condition links the choice of the penalty parameter to the gradient of the objective function.
**Condition 1.** *The $\ell_1$-penalty parameter strongly enforces regularization if it is greater than any partial derivatives of the objective function at $\theta_u = \theta_u^*$, i.e.*

$$\lambda \geq 2 \left\|\nabla \mathcal{S}_n\left(\theta_u^*\right)\right\|_{\infty}. \tag{13}$$

Condition 1 guarantees that if the vector of couplings $\theta_u^*$ has at most $d$ non-zero entries, then the estimation difference $\widehat{\underline{\theta}}_u\left(\lambda\right) - \underline{\theta}_u^*$ lies within the set

$$K := \left\{\Delta \in \mathbb{R}^{p-1} \mid \left\|\Delta\right\|_1 \leq 4\sqrt{d} \left\|\Delta\right\|_2\right\}. \tag{14}$$

The second condition ensure that the objective function is strongly convex in a restricted subset of $\mathbb{R}^{p-1}$. Denote the reminder of the first-order Taylor expansion of the objective function by

$$\delta \mathcal{S}_n\left(\Delta_u, \theta_u^*\right) := \mathcal{S}_n\left(\theta_u^* + \Delta_u\right) - \mathcal{S}_n\left(\theta_u^*\right) - \left\langle\nabla \mathcal{S}_n\left(\theta_u^*\right), \Delta_u\right\rangle, \tag{15}$$

where $\Delta_u \in \mathbb{R}^{p-1}$ is an arbitrary vector. Then the second condition reads as follows.
**Condition 2.** *The objective function is restricted strongly convex with respect to $K$ on a ball of radius $R$ centered at $\theta_u = \theta_u^*$, if for all $\Delta_u \in K$ such that $\left\|\Delta_u\right\|_2 \leq R$, there exists a constant $\kappa > 0$ such that*

$$\delta \mathcal{S}_n\left(\Delta_u, \theta_u^*\right) \geq \kappa \left\|\Delta_u\right\|_2^2. \tag{16}$$

*Strong regularization and restricted strong convexity enables us to control that the minimizer $\widehat{\underline{\theta}}_u$ of the full objective (9) lies in the vicinity of the sparse vector of parameters $\theta_u^*$. The precise formulation is given in the proposition following from [22, Thm. 1].*
**Proposition 1.** *If the $\ell_1$-regularized M-estimator of the form (9) satisfies Condition 1 and Condition 2 with $R > 3\sqrt{d}\frac{\lambda}{\kappa}$ then the square-error is bounded by*

$$\left\|\widehat{\underline{\theta}}_u - \underline{\theta}_u^*\right\|_2 \leq 3\sqrt{d}\frac{\lambda}{\kappa}. \tag{17}$$

## 3.1 Gradient Concentration

Like the ISO (8), its gradient in any component $l \in V \setminus u$ is an empirical average

$$\frac{\partial}{\partial \theta_{ul}} \mathcal{S}_n \left( \underline{\theta}_u \right) = \frac{1}{n} \sum_{k=1}^{n} X_{ul}^{(k)} \left( \underline{\theta}_u \right), \tag{18}$$

where the random variables $X_{ul}^{(k)} \left( \underline{\theta}_u \right)$ are i.i.d and they are related to the spin configurations according to

$$X_{ul} \left( \underline{\theta}_u \right) = -\sigma_u \sigma_l \exp \left( - \sum_{i \in V \setminus u} \theta_{ui} \sigma_u \sigma_i \right). \tag{19}$$

In order to prove that the ISO gradient concentrates we have to state few properties of the support, the mean and the variance of the random variables (19), expressed in the following three Lemmas.

The first of the Lemmas states that at $\underline{\theta}_u = \underline{\theta}_u^*$, the random variable $X_{ul} \left( \underline{\theta}_u^* \right)$ has zero mean.

**Lemma 1.** *For any Ising model with $p$ spins and for all $l \neq u \in V$*

$$\mathbb{E} \left[ X_{ul} \left( \underline{\theta}_u^* \right) \right] = 0. \tag{20}$$

As a direct corollary of the Lemma 1, $\underline{\theta}_u = \underline{\theta}_u^*$ is always a minimum of the averaged ISO (8).

The second Lemma proves that at $\underline{\theta}_u = \underline{\theta}_u^*$, the random variable $X_{ul} \left( \underline{\theta}_u^* \right)$ has a variance equal to one.

**Lemma 2.** *For any Ising model with $p$ spins and for all $l \neq u \in V$*

$$\mathbb{E} \left[ X_{ul} \left( \underline{\theta}_u^* \right)^2 \right] = 1. \tag{21}$$

The next lemma states that at $\underline{\theta}_u = \underline{\theta}_u^*$, the random variable $X_{ul} \left( \underline{\theta}_u^* \right)$ has a bounded support.

**Lemma 3.** *For any Ising model with $p$ spins, with maximum degree $d$ and maximum coupling intensity $\beta$, it is guaranteed that for all $l \neq u \in V$*

$$\left| X_{ul} \left( \underline{\theta}_u^* \right) \right| \leq \exp \left( \beta d \right). \tag{22}$$

With Lemma 1, 2 and 3, and using Berstein's inequality we are now in position to prove that every partial derivative of the ISO concentrates uniformly around zero as the number of samples grows.

**Lemma 4.** *For any Ising model with $p$ spins, with maximum degree $d$ and maximum coupling intensity $\beta$. For any $\epsilon_3 > 0$, if the number of observation satisfies $n \geq \exp \left( 2\beta d \right) \ln \frac{2p}{\epsilon_3}$, then the following bound holds with probability at least $1 - \epsilon_3$:*

$$\left\| \nabla \mathcal{S}_n \left( \underline{\theta}_u^* \right) \right\|_\infty \leq 2 \sqrt{\frac{\ln \frac{2p}{\epsilon_3}}{n}}. \tag{23}$$

## 3.2 Restricted Strong-Convexity

The remainder of the first-order Taylor-expansion of the ISO, defined in Eq. (15) is explicitly computed

$$\delta \mathcal{S}_n \left( \Delta_u, \theta^* \right) = \frac{1}{n} \sum_{k=1}^{n} \exp \left( - \sum_{i \in \partial u} \theta_{ui}^* \sigma_u^{(k)} \sigma_i^{(k)} \right) f \left( \sum_{i \in V \setminus u} \Delta_{ui} \sigma_u^{(k)} \sigma_i^{(k)} \right), \tag{24}$$

where $f(z) := e^{-z} - 1 + z$.

In the following lemma we prove that Eq. (24) is controlled by a much simpler expression using a lower-bound on $f(z)$.

**Lemma 5.** *For all $\Delta_u \in \mathbb{R}^{p-1}$, the remainder of the first-order Taylor expansion admits the following lower-bound*

$$\delta \mathcal{S}_n \left( \Delta_u, \theta^* \right) \geq \frac{e^{-\beta d}}{2 + \left\| \Delta_u \right\|_1} \Delta_u^\top H^n \Delta_u \tag{25}$$

*where $H^n$ is an empirical covariance matrix with elements $H_{ij}^n = \frac{1}{n} \sum_{k=1}^{n} \sigma_i^{(k)} \sigma_j^{(k)}$ for $i, j \in V \setminus u$.*

Lemma 5 enables us to control the randomness in $\delta\mathcal{S}_n(\Delta_u, \theta^*)$ through the simpler matrix $H^n$ that is *independent* of $\Delta_u$. This last point is crucial as we show in the next lemma that $H^n$ concentrates independently of $\Delta_u$ towards its mean.

**Lemma 6.** *Consider an Ising model with $p$ spins, with maximum degree $d$ and maximum coupling intensity $\beta$. Let $\delta > 0$, $\epsilon_4 > 0$ and $n \geq \frac{2}{\delta^2} \ln \frac{p^2}{\epsilon_4}$. Then with probability greater than $1 - \epsilon_4$, we have for all $i, j \in V \setminus u$*

$$\left| H_{ij}^n - H_{ij} \right| \leq \delta, \tag{26}$$

*where the matrix $H$ is the covariance matrix with elements $H_{ij} = \mathbb{E}\left[\sigma_i \sigma_j\right]$, for $i, j \in V \setminus u$.*

The last ingredient that we need is a proof that the smallest eigenvalue of the covariance matrix $H$ is bounded away from zero *independently of the dimension $p$*. Equivalently the next lemma shows that the quadratic form associated with $H$ is non-degenerate regardless of the value of $p$.

**Lemma 7.** *Consider an Ising model with $p$ spins, with maximum degree $d$ and maximum coupling intensity $\beta$. For all $\Delta_u \in \mathbb{R}^{p-1}$ the following bound holds*

$$\Delta_u^\top H \Delta_u \geq \frac{e^{-2\beta d}}{d+1} \left\| \Delta_u \right\|_2^2. \tag{27}$$

We stress that Lemma 7 is a deterministic result valid for all $\Delta_u \in \mathbb{R}^{p-1}$. We are now in position to prove the restricted strong convexity of the ISO.

**Lemma 8.** *Consider an Ising model with $p$ spins, with maximum degree $d$ and maximum coupling intensity $\beta$. For all $\epsilon_4 > 0$ and $R > 0$, when $n \geq 2^{11} d^2 (d+1)^2 e^{4\beta d} \ln \frac{p^2}{\epsilon_4}$ the ISO (8) satisfies, with probability at least $1 - \epsilon_4$, the restricted strong convexity condition*

$$\delta\mathcal{S}_n\left(\Delta_u, \theta_u^*\right) \geq \frac{e^{-3\beta d}}{4(d+1)\left(1 + 2\sqrt{d}R\right)} \left\| \Delta_u \right\|_2^2, \tag{28}$$

*for all $\Delta_u \in \mathbb{R}^{p-1}$ such that $\left\| \Delta_u \right\|_1 \leq 4\sqrt{d} \left\| \Delta_u \right\|_2$ and $\left\| \Delta_u \right\|_2 \leq R$.*

### 3.3 Proof of the main Theorems

*Proof of Theorem 1 (Square Error of RISE).* We seek to apply Proposition 1 to the Regularized Interaction Screening Estimator (9). Using $\epsilon_3 = \frac{2\epsilon_1}{3}$ in Lemma 4 and letting $\lambda = 4\sqrt{\frac{1}{n} \ln 3p/\epsilon_1}$, it follows that Condition 1 is satisfied with probability greater than $1 - 2\epsilon_1/3$, whenever $n \geq e^{2\beta d} \ln \frac{3p}{\epsilon_1}$. Using $\epsilon_4 = \epsilon_1/3$ in Lemma 8, and observing that $12\sqrt{d}\lambda e^{3\beta d}(d+1)(1 + 2\sqrt{d}R) < R$, for $R = 2/\sqrt{d}$ and $n \geq 2^{14} d^2 (d+1)^2 e^{6\beta d} \ln \frac{3p^2}{\epsilon_1}$, we conclude that condition 2 is satisfied with probability greater than $1 - \frac{\epsilon_1}{3}$. Theorem 1 then follows by using a union bound and then applying Proposition 1. $\square$

The proof of Theorem 2 becomes an immediate application of Theorem 1 for achieving an estimation of couplings at each node with squared-error of $\alpha/2$ and with probability $1 - \epsilon_1 = 1 - \epsilon_2/p$.

## 4 Numerical Results

We test performance of the Struct-RISE, with the strength of the $l_1$-regularization parametrized by $\lambda = 4\sqrt{\frac{1}{n} \ln(3p^2/\epsilon)}$, on Ising models over two-dimensional grid with periodic boundary conditions (thus degree of every node in the graph is 4). We have observed that this topology is one of the hardest for the reconstruction problem. We are interested to find the minimal number of samples, $n_{\min}$, such that the graph is perfectly reconstructed with probability $1 - \epsilon \geq 0.95$. In our numerical experiments, we recover the value of $n_{\min}$ as the minimal $n$ for which Struct-RISE outputs the perfect structure 45 times from 45 different trials with $n$ samples, thus guaranteeing that the probability of perfect reconstruction is greater than 0.95 with a statistical confidence of at least 90%.

We first verify the logarithmic scaling of $n_{\min}$ with respect to the number of spins $p$. The couplings are chosen uniform and positive $\theta_{ij}^* = 0.7$. This choice ensures that samples generated by Glauber

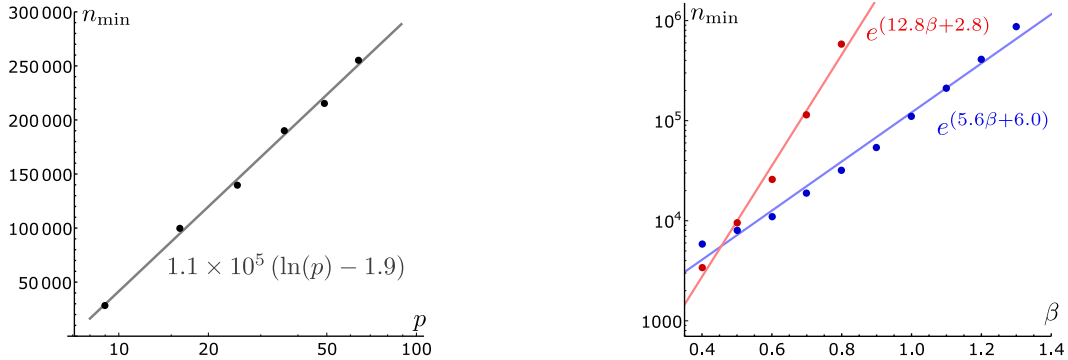

Figure 1: Left: Linear-exponential plot showing the observed relation between $n_{\min}$ and $p$. The graph is a $\sqrt{p} \times \sqrt{p}$ two-dimensional grid with uniform and positive couplings $\theta^* = 0.7$. Right: Linear-exponential plot showing the observed relation between $n_{\min}$ and $\beta$. The graph is the two-dimensional $4 \times 4$ grid. In red the couplings are uniform and positive and in blue the couplings have uniform intensity but random sign.

dynamics are i.i.d. according to (1). Values of $n_{\min}$ for $p \in \{9, 16, 25, 36, 49, 64\}$ are shown on the left in Figure 1. Empirical scaling is, $\approx 1.1 \times 10^5 \ln p$, which is orders of magnitude better than the rather conservative prediction of the theory for this model, $3.2 \times 10^{15} \ln p$.

We also test the exponential scaling of $n_{\min}$ with respect to the maximum coupling intensity $\beta$. The test is conducted over two different settings both with $p = 16$ spins: the ferromagnetic case where all couplings are uniform and positive, and the spin glass case where the sign of couplings is assigned uniformly at random. In both cases the absolute value of the couplings, $\left| \theta_{ij}^* \right|$, is uniform and equal to $\beta$. To ensure that the samples are i.i.d, we sample directly from the exhaustive weighted list of the $2^{16}$ possible spin configurations. The structure is recovered by thresholding the reconstructed couplings at the value $\alpha/2 = \beta/2$.

Experimental values of $n_{\min}$ for different values of the maximum coupling intensity, $\beta$, are shown on the right in Fig. 1. Empirically observed exponential dependence on $\beta$ is matched best by, $\exp(12.8\beta)$, in the ferromagnetic case and by, $\exp(5.6\beta)$, in the case of the spin glass. Theoretical bound for $d = 4$ predicts $\exp(24\beta)$. We observe that the difference in sample complexity depends significantly on the type of interaction. An interesting observation one can make based on these experiments is that the case which is harder from the sample-generating perspective is easier for learning and vice versa.

## 5 Conclusions and Path Forward

In this paper we construct and analyze the Regularized Interaction Screening Estimator (9). We show that the estimator is computationally efficient and needs an optimal number of samples for learning Ising models. The RISE estimator does not require any prior knowledge about the model parameters for implementation and it is based on the minimization of the loss function (8), that we call the Interaction Screening Objective. The ISO is an empirical average (over samples) of an objective designed to screen an individual spin/variable from its factor-graph neighbors.

Even though we focus in this paper solely on learning pair-wise binary models, the "interaction screening" approach we introduce here is generic. The approach extends to learning other Graphical Models, including those over higher (discrete, continuous or mixed) alphabets and involving high-order (beyond pair-wise) interactions. These generalizations are built around the same basic idea pioneered in this paper – the interaction screening objective is (a) minimized over candidate GM parameters at the actual values of the parameters we aim to learn; and (b) it is an empirical average over samples. In the future, we plan to explore further theoretical and experimental power, characteristics and performance of the generalized screening estimator.

**Acknowledgment**: We are thankful to Guy Bresler and Andrea Montanari for valuable discussions, comments and insights. The work was supported by funding from the U.S. Department of Energy's Office of Electricity as part of the DOE Grid Modernization Initiative.

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
