[Supplementary Material]

# Supplementary Materials for Interaction Screening: Efficient and Sample-Optimal Learning of Ising Models

**Marc Vuffray**[1], **Sidhant Misra**[2], **Andrey Y. Lokhov**[1,3], and **Michael Chertkov**[1,3,4]

[1]Theoretical Division T-4, Los Alamos National Laboratory, Los Alamos, NM 87545, USA
[2]Theoretical Division T-5, Los Alamos National Laboratory, Los Alamos, NM 87545, USA
[3]Center for Nonlinear Studies, Los Alamos National Laboratory, Los Alamos, NM 87545, USA
[4]Skolkovo Institute of Science and Technology, 143026 Moscow, Russia

{vuffray, sidhant, lokhov, chertkov}@lanl.gov

We provide in Section 1 detailed proofs of Lemma 1, 2, 3 and 4 related to the gradient concentration of the Interaction-Screening Objective (ISO). Detailed proofs of Lemma 5, 6, 7 and 8 related to the restricted strong-convexity of the ISO can be found in Section 2.

## 1 Gradient Concentration

**Lemma 1.** *For any Ising model with $p$ spins and for all $l \neq u \in V$*

$$\mathbb{E}\left[X_{ul}\left(\underline{\theta}_u^*\right)\right] = 0. \tag{1}$$

*Proof.* By direct computation, we find that

$$
\begin{aligned}
\mathbb{E}\left[X_{ul}\left(\underline{\theta}_u^*\right)\right] &= \mathbb{E}\left[-\sigma_u\sigma_l\exp\left(-\sum_{i\in\partial u}\theta_{ui}^*\sigma_u\sigma_i\right)\right] \\
&= \frac{-1}{Z}\sum_{\underline{\sigma}}\sigma_u\sigma_l\exp\left(\sum_{(i,j)\in E}\theta_{ij}^*\sigma_i\sigma_j - \sum_{i\in\partial u}\theta_{ui}^*\sigma_u\sigma_i\right) = 0,
\end{aligned}
\tag{2}
$$

where in the last line we use the fact that the exponential terms involving $\sigma_u$ cancel, implying that the sum over $\sigma_u \in \{-1,+1\}$ is zero. $\qquad\square$

**Lemma 2.** *For any Ising model with $p$ spins and for all $l \neq u \in V$*

$$\mathbb{E}\left[X_{ul}\left(\underline{\theta}_u^*\right)^2\right] = 1. \tag{3}$$

*Proof.* As a result of direct evaluation one derives

$$
\begin{aligned}
\mathbb{E}\left[X_{ul}\left(\underline{\theta}_u^*\right)^2\right] &= \mathbb{E}\left[\exp\left(-2\sum_{i\in\partial u}\theta_{ui}^*\sigma_u\sigma_i\right)\right] \\
&= \frac{1}{Z}\sum_{\underline{\sigma}}\exp\left(\sum_{(i,j)\in E,i,j\neq u}\theta_{ij}^*\sigma_i\sigma_j - \sum_{i\in\partial u}\theta_{ui}^*\sigma_u\sigma_i\right) \\
&= \frac{1}{Z}\sum_{\underline{\sigma}}\exp\left(\sum_{(i,j)\in E,i,j\neq u}\theta_{ij}^*\sigma_i\sigma_j + \sum_{i\in\partial u}\theta_{ui}^*\sigma_u\sigma_i\right) \\
&= 1.
\end{aligned}
\tag{4}
$$

Notice that in the second line the first sum over edges (under the exponential) does not depend on $\sigma_u$. Furthermore, the first sum is invariant under the change of variables, $\sigma_u \to -\sigma_u$, while the second sum changes sign. This transformation results in appearance of the partition function in the numerator. $\square$

**Lemma 3.** *For any Ising model with $p$ spins, with maximum degree $d$ and maximum coupling intensity $\beta$, it is guaranteed that for all $l \neq u \in V$*

$$
|X_{ul}\left(\underline{\theta}_u^*\right)| \leq \exp\left(\beta d\right).
\tag{5}
$$

*Proof.* Observe that components of $\underline{\theta}_u^*$ are smaller than $\beta$ and at most $d$ of them are non-zero. Recall that spins are binary, $\{-1,+1\}$, which results in the following estimate

$$
\begin{aligned}
|X_{ul}\left(\underline{\theta}_u^*\right)| &= \left|-\sigma_u\sigma_i\exp\left(-\sum_{i\in\partial u}\theta_{ui}^*\sigma_u\sigma_i\right)\right| \\
&\leq \exp\left(-\sum_{i\in\partial u}\theta_{ui}^*\sigma_u\sigma_i\right) \\
&\leq \exp\left(\beta d\right).
\end{aligned}
\tag{6}
$$

$\square$

**Lemma 4.** *For any Ising model with $p$ spins, with maximum degree $d$ and maximum coupling intensity $\beta$. For any $\epsilon_3 > 0$, if the number of observation satisfies $n \geq \exp\left(2\beta d\right)\ln\frac{2p}{\epsilon_3}$, then the following bound holds with probability at least $1 - \epsilon_3$:*

$$
\|\nabla\mathcal{S}_n\left(\underline{\theta}_u^*\right)\|_\infty \leq 2\sqrt{\frac{\ln\frac{2p}{\epsilon_3}}{n}}.
\tag{7}
$$

*Proof.* Let us first show that every term is individually bounded by the RHS of (7) with high-probability. We further use the union bound to prove that all components are uniformly bounded with high-probability. Utilizing Lemma 1, Lemma 2 and Lemma 3 we apply the Bernstein's Inequality

$$
\mathbb{P}\left[\left|\frac{\partial}{\partial\theta_{ul}}\mathcal{S}_n\left(\underline{\theta}_u^*\right)\right| > t\right] \leq 2\exp\left(-\frac{\frac{1}{2}t^2 n}{1 + \frac{1}{3}\exp\left(\beta d\right)t}\right).
\tag{8}
$$

Inverting the following relation

$$
s = \frac{\frac{1}{2}t^2 n}{1 + \frac{1}{3}\exp\left(\beta d\right)t},
\tag{9}
$$

and substituting the result in the Eq. (8) one derives

$$
\mathbb{P}\left[\left|\frac{\partial}{\partial\theta_{ul}}\mathcal{S}_n\left(\underline{\theta}_u^*\right)\right| > \frac{1}{3}\left(u + \sqrt{\frac{18}{\exp\left(\beta d\right)}u + u^2}\right)\right] \leq 2\exp\left(-s\right),
\tag{10}
$$

where $u = \frac{s}{n}\exp\left(\beta d\right)$.

For $n \geq s \exp(2\beta d)$, we can simplify Eq. (10) to have an expression independent of $\beta$ and $d$

$$\mathbb{P}\left[\left|\frac{\partial}{\partial \theta_{ul}} \mathcal{S}_n(\theta_u^*)\right| > 2\sqrt{\frac{s}{n}}\right] \leq 2\exp(-s). \tag{11}$$

Using $s = \ln \frac{2p}{\epsilon_3}$ and the union bound on every component of the gradient leads to the desired result. $\qquad \square$

## 2 Restricted Strong-Convexity

We recall that the remainder of the first-order Taylor-expansion of the ISO, is the following quantity

$$\delta\mathcal{S}_n(\Delta_u, \theta^*) = \frac{1}{n}\sum_{k=1}^n \exp\left(-\sum_{i \in \partial u} \theta_{ui}^* \sigma_u^{(k)}\sigma_i^{(k)}\right) f\left(\sum_{i \in V\backslash u} \Delta_{ui}\sigma_u^{(k)}\sigma_i^{(k)}\right), \tag{12}$$

where the function $f(z)$ appearing in Eq. (12) reads

$$f(z) := e^{-z} - 1 + z. \tag{13}$$

**Lemma 5.** *For all $\Delta_u \in \mathbb{R}^{p-1}$, the remainder of the first-order Taylor expansion admits the following lower-bound*

$$\delta\mathcal{S}_n(\Delta_u, \theta^*) \geq \frac{e^{-\beta d}}{2 + \|\Delta_u\|_1} \Delta_u^\top H^n \Delta_u \tag{14}$$

*where the matrix $H^n$ is an empirical covariance matrix with elements $i, j \in V\backslash u$*

$$H_{ij}^n = \frac{1}{n}\sum_{k=1}^n \sigma_i^{(k)}\sigma_j^{(k)}. \tag{15}$$

*Proof.* We start to prove a lower-bound on the function $f(z)$ valid for all $z \in \mathbb{R}$,

$$f(z) \geq \frac{z^2}{2 + |z|}. \tag{16}$$

To see this, define an auxiliary function $g(z)$ as follows

$$\begin{aligned} g(z) :&= (2 + |z|)f(z) - z^2 \\ &= (2 + |z|)(e^{-z} - 1 + z) - z^2. \end{aligned} \tag{17}$$

We show that $g(z)$ achieves its minimum at $g(0) = 0$. Observe that the first derivative of $g(z)$ vanishes at zero from both the negative and positive side

$$\lim_{z \to 0_+} \frac{d}{dz}g(z) = \lim_{z \to 0_-} \frac{d}{dz}g(z) = 0. \tag{18}$$

Moreover for all $z > 0$ the second derivative of $g(z)$ is non-negative

$$\frac{d^2}{dz^2}g(z) = ze^{-z} > 0. \tag{19}$$

A similar result holds for $z < 0$

$$\frac{d^2}{dz^2}g(z) = 4(e^{-z} - 1) - ze^{-z} > 0, \tag{20}$$

proving that for all $z$, $g(z) \geq g(0) = 0$.

Combining Eq. (16) with the straightforward inequalities

$$\left|\sum_{i \in V\backslash u} \Delta_{ui}\sigma_u^{(k)}\sigma_i^{(k)}\right| \leq \|\Delta_u\|_1, \tag{21}$$

and

$$\exp\left(-\sum_{i\in\partial u}\theta_{ui}^{*}\sigma_{u}^{(k)}\sigma_{i}^{(k)}\right)\geq\exp\left(-\beta d\right),\tag{22}$$

leads us to the following lower-bound on the remainder of the first-order Taylor expansion of the ISO

$$\delta\mathcal{S}_{n}\left(\Delta_{u},\theta^{*}\right)\geq\frac{e^{-\beta d}}{2+\|\Delta_{u}\|_{1}}\frac{1}{n}\sum_{k=1}^{n}\left(\sum_{i\in V\setminus u}\Delta_{ui}\sigma_{u}^{(k)}\sigma_{i}^{(k)}\right)^{2}$$

$$=\frac{e^{-\beta d}}{2+\|\Delta_{u}\|_{1}}\Delta_{u}^{\top}H^{n}\Delta_{u},\tag{23}$$

where in the last line we used the trivial identity $\sigma_{u}^{(k)}\cdot\sigma_{u}^{(k)}=1$. $\qquad\square$

**Lemma 6.** *Consider an Ising model with $p$ spins, with maximum degree $d$ and maximum coupling intensity $\beta$. Let $\delta > 0$, $\epsilon_4 > 0$ and $n \geq \frac{2}{\delta^2}\ln\frac{p^2}{\epsilon_4}$. Then with probability greater than $1 - \epsilon_4$, we have for all $i, j \in V \setminus u$*

$$\left|H_{ij}^{n}-H_{ij}\right|\leq\delta,\tag{24}$$

*where the matrix $H$ is the covariance matrix with elements $i, j \in V \setminus u$*

$$H_{ij}=\mathbb{E}\left[\sigma_{i}\sigma_{j}\right].\tag{25}$$

*Proof.* We recall that the matrix elements of the empirical covariance matrix read

$$H_{ij}^{n}=\frac{1}{n}\sum_{k=1}^{n}\sigma_{i}^{(k)}\sigma_{j}^{(k)}.\tag{26}$$

Since $\left|\sigma_{i}^{(k)}\sigma_{j}^{(k)}\right|\leq 1$ using Hoeffding's inequality, we have

$$\mathbb{P}\left[\left|H_{ij}^{n}-H_{ij}\right|\geq\delta\right]\leq 2\exp\left(-\frac{n\delta^{2}}{2}\right).\tag{27}$$

As $H_{ij}^{n}$ is symmetric we use the union bound over the elements $i < j \in V \setminus u$ to get

$$\mathbb{P}\left[\left|H_{ij}^{n}-H_{ij}\right|\geq\delta\quad\forall i,j\in V\setminus u\right]\leq 1-p^{2}\exp\left(-\frac{n\delta^{2}}{2}\right).\tag{28}$$

$\qquad\square$

**Lemma 7.** *Consider an Ising model with $p$ spins, with maximum degree $d$ and maximum coupling intensity $\beta$. For all $\Delta_u \in \mathbb{R}^{p-1}$ the following bound holds*

$$\Delta_{u}^{\top}H\Delta_{u}\geq\frac{e^{-2\beta d}}{d+1}\|\Delta_{u}\|_{2}^{2}.\tag{29}$$

*Proof.* Our proof strategy here follows [1, Cor. 3.1]. Notice that the probability measure of the Ising model is symmetric with respect to the sign flip, i.e. $\mu\left(\sigma_1,\ldots,\sigma_p\right)=\mu\left(-\sigma_1,\ldots,-\sigma_p\right)$. Thus any spin has zero mean, which implies that for every $\Delta_u \in \mathbb{R}^{p-1}$

$$\mathbb{E}\left[\left(\sum_{i\in V\setminus u}\Delta_{ui}\sigma_{i}\right)\right]=0.\tag{30}$$

This allows to reinterpret the left-hand side of Eq. (29) as a variance, using that $\sigma_u^2 = 1$,

$$\Delta_{u}^{\top}H\Delta_{u}=\sum_{i,j\in V\setminus u}\Delta_{ui}\mathbb{E}\left[\sigma_{i}\sigma_{j}\right]\Delta_{uj}$$

$$=\mathbb{E}\left[\left(\sum_{i\in V\setminus u}\Delta_{ui}\sigma_{i}\right)^{2}\right]$$

$$=\mathrm{Var}\left[\sum_{i\in V\setminus u}\Delta_{ui}\sigma_{i}\right].\tag{31}$$

Construct a subset $A \subset V$ recursively as follows: (i) let $i_0 = \text{argmax}_{j \in V \setminus u} \Delta_{uj}^2$ and define $A_0 = \{i_0\}$, (ii) given $A_t = \{i_0, \ldots, i_t\}$, let $B_t = \{j \in V \setminus A_t \mid \partial j \cap A_t = \emptyset\}$ and $i_{t+1} = \text{argmax}_{j \in B_t \setminus u} \Delta_{uj}^2$ and set $A_{t+1} = A_t \cup \{i_{t+1}\}$, (iii) terminate when $B_t \setminus u = \emptyset$ and declare $A = A_t$.

The set $A$ possesses the following two main properties. First, every node $i \in A$ does not have any neighbors in $A$ and, second,

$$(d+1) \sum_{i \in A} \Delta_{ui}^2 \geq \sum_{i \in V \setminus u} \Delta_{ui}^2. \tag{32}$$

We apply the law of total variance to (31) by conditioning on the set of spins $\underline{\sigma}_{A^c}$ with indexes belonging to the complementary set $A^c$,

$$\text{Var}\left[\sum_{i \in V \setminus u} \Delta_{ui}\sigma_i\right] \geq \mathbb{E}\left[\text{Var}\left[\sum_{i \in V \setminus u} \Delta_{ui}\sigma_i \;\middle|\; \underline{\sigma}_{A^c}\right]\right]$$

$$= \sum_{i \in A} \Delta_{ui}^2 \mathbb{E}\left[\text{Var}\left[\sigma_i \mid \underline{\sigma}_{A^c}\right]\right], \tag{33}$$

where in the last line one uses that the spins in $A$ are conditionally independent given their neighbors $\underline{\sigma}_{A^c}$. One concludes the proof by using relation (32) and the fact that the conditional variance of a spin given its neighbors is bounded from below:

$$\text{Var}\left[\sigma_i \mid \underline{\sigma}_{A^c}\right] = 1 - \tanh^2\left(\sum_{j \in \partial i} \theta_{ij}^* \sigma_j\right)$$

$$\geq \exp(-2\beta d). \tag{34}$$

$\square$

**Lemma 8.** *Consider an Ising model with $p$ spins, with maximum degree $d$ and maximum coupling intensity $\beta$. For all $\epsilon_4 > 0$ and $R > 0$, when $n \geq 2^{11} d^2 (d+1)^2 e^{4\beta d} \ln \frac{p^2}{\epsilon_4}$ the ISO satisfies, with probability at least $1 - \epsilon_4$, the restricted strong convexity condition*

$$\delta\mathcal{S}_n(\Delta_u, \theta_u^*) \geq \frac{e^{-3\beta d}}{4(d+1)\left(1 + 2\sqrt{d}R\right)} \|\Delta_u\|_2^2, \tag{35}$$

*for all $\Delta_u \in \mathbb{R}^{p-1}$ such that $\|\Delta_u\|_1 \leq 4\sqrt{d}\|\Delta_u\|_2$ and $\|\Delta_u\|_2 \leq R$.*

*Proof.* First we apply Lemma 5 to get the quadratic bound

$$\delta\mathcal{S}_n(\Delta_u, \theta^*) \geq \frac{e^{-\beta d}}{2 + \|\Delta_u\|_1} \Delta_u^\top H^n \Delta_u$$

$$\geq \frac{e^{-\beta d}}{2\left(1 + 2\sqrt{d}R\right)} \Delta_u^\top H^n \Delta_u. \tag{36}$$

Second we use Lemma 7 to bound the quadratic form

$$\Delta_u^\top H^n \Delta_u = \Delta_u^\top H \Delta_u + \Delta_u^\top (H^n - H) \Delta_u$$

$$\geq \frac{e^{-2\beta d}}{d+1} \|\Delta_u\|_2^2 + \Delta_u^\top (H^n - H) \Delta_u. \tag{37}$$

Third we conclude with Lemma 6, controlling randomness independently of $\Delta_u$. Choosing $\delta = \frac{e^{-2\beta d}}{32d(d+1)}$, we get with probability at least $1 - \epsilon_4$ that

$$\Delta_u^\top (H^n - H) \Delta_u \geq -\frac{e^{-2\beta d}}{32d(d+1)} \|\Delta_u\|_1^2$$

$$\geq -\frac{e^{-2\beta d}}{2(d+1)} \|\Delta_u\|_2^2, \tag{38}$$

whenever $n \geq \frac{2}{\delta^2} \ln \frac{p^2}{\epsilon_4} = 2^{11} d^2 (d+1)^2 e^{4\beta d} \ln \frac{p^2}{\epsilon_4}$. $\square$

# References

[1] A. Montanari, "Computational implications of reducing data to sufficient statistics," *Electron. J. Statist.*, vol. 9, no. 2, pp. 2370–2390, 2015.