[Reviews · NeurIPS 2016]

Reviewer 1

Summary

The paper studies the problem of learning bounded degree Ising models, with p nodes, and a,b, and d denote the lower and upper bounds on edge strengths, and d is the bound on the degree. They consider the sample and computation complexity of learning such Ising models. The previous known best results were by Bresler, requiring O(log p*exp(exp(bd))) samples and time p^2. I am suppressing the dependence on a. The major drawback of the algorithm is the requirement of doubly exponential number of samples as a function of d. The know lower bounds are singly exponential, and there are inefficient algorithms achieving this. This paper removes the doubly exponential dependence, and achieves a nearly optimal bound. The computation required is however O(p^3), which the authors believe might be reducable to O(p^2) (lines 169-170). This is a little odd given that they claim quasi-quadratic time in lines 62-63. I am not sure if they indeed have such an algorithm. One other nice thing about the paper is that they do not need to know beta, or d apriori.

Qualitative Assessment

I really enjoyed reading the paper. It provides a clean convex optimization formulation of Ising model learning problem. This results in an exponential improvement over the previous known sample complexity results. L 63: Please clarify if indeed there is a quasi-quadratic time algorithm. vertexes -> vertices L176: Naghaband -> Negahban

Confidence in this Review

2-Confident (read it all; understood it all reasonably well)


Reviewer 2

Summary

The paper considers the problem of inferring the structure of Ising models from i.i.d. samples and proposes a novel estimator that is the optimum of a regularized scoring function for each variable. The optima of this function indicate which edges are in the true underlying graph structure with high probability.

Qualitative Assessment

What is remarkable about this paper is how superficially similar the proposed objective is to the regularized pseudolikelihood objective that is also used for structure learning, but how much weaker the proved sufficient conditions are for recovering the true graph with high probability. As the paper argues, those methods require assumptions on spectral norms of inverted parameter matrices, essentially meaning that the true structure isn't too ambiguous due to strong couplings, which is hard to verify from the given samples. The conditions presented here are more user friendly. The fact that it is closer to sample optimal is nice plus. Questions: The intuition behind the estimator is still a bit unclear to me. Can the authors expand on what it means for one variable to screen another? What method did you use to optimize the RISE objective? How computationally expensive was it with respect to wall clock time? I like the suggestion of improving speed with gradient-based methods.

Confidence in this Review

2-Confident (read it all; understood it all reasonably well)


Reviewer 3

Summary

This paper proposes an algorithm for inferring the network structure of Ising model from spin observations. The algorithm is computationally efficient, and it achieves the optimal rate (an information theoretic lower-bound) of required sample size for recovering the network. The setting is similar to [21], and the evaluation is based on the theory of [22].

Qualitative Assessment

The information theoretic lower-bound is often conservative. In fact, the numerical examples of Figure 1 show that the theoretically predicted orders are extremely larger than the experimentally observed orders (see line 259, and line 269). This may imply a practical limitation of the theory for evaluating estimation methods, and statistical efficiency, say, often gives better understanding. The idea of the proposed algorithm is to decompose the model (1) into the node-wise terms, and consider the optimization separately for each node. Similar idea is found in the well-known method for GGM with lasso (Meinshausen and Buhlmann 2006; this should be cited?). However, the particular loss-function (Eq. (8)) proposed in this paper is unique and very intriguing. In statistical literature, the loss function often takes the form of –log p for the partial probability model p. But (8) is unique in the sense that it takes the form of 1/p, and it leads to the interesting properties shown in Lemma 1 and Lemma 2. So, the estimation algorithm with this interesting form of loss function would be worth to be noted. Typo: the axis is beta, not p, in the right panel of Figure 1.

Confidence in this Review

2-Confident (read it all; understood it all reasonably well)


Reviewer 4

Summary

The paper proposes a new estimator based on the loss function called Interaction Screening Objective (ISO). The authors add L1 regularization to the loss function to reconstruct the structure of Ising model with an error probability. One of the advantages of their estimator is no need prior knowledge on the graph parameters. They also demonstrate their structure-learning algorithm is low complexity with respect to the information- theoretic lower bound.

Qualitative Assessment

1. In the reference [17] of the paper, the loss function can be explained as the conditional probability of one vertex given the others. Can the authors give some more intuition for using Interaction Screening Objective (ISO) loss function? 2. At the line 135, can the authors offer a simple example to explain the “independent of the rest”? 3. In the right subfigure of Figure 1, should p be beta? Can the authors provide the case that the absolute value of the couplings is not the same?

Confidence in this Review

2-Confident (read it all; understood it all reasonably well)


Reviewer 5

Summary

The authors proposed a new method(ISO) to estimate the Ising model. They use the idea of neighborhood selection to estimate the Ising model. For each node u, their l_1 penalized ISO can achieve l2 consistency. And they proved their estimator is model consistent under much milder assumptions with appropriate regularization. The sample number needed to get such a consistent estimate is near-optimal w.r.t information theoretic lower-bound. Also the experiments validates the rate of needed sample number.

Qualitative Assessment

The paper is technical sound and well written. The proof basic follows the classical method based on the RSC condition which seems all right. I have only two suggestions: 1. For neighborhood selection, the estimate is usually not symmetric. So why not add up all the ISO for nodes, and use the summation as a big loss function. Such a loss function can induce a symmetric estimate. Will RSC condition still hold. 2. In the experiment part, they should compare this new method to the existed method, for example ref[17]. I believe in easy setting [17] can out-perform this new method, but in some singular setting, ISO may be better. Such a comparison is helpful to understand the difference between these two method.

Confidence in this Review

2-Confident (read it all; understood it all reasonably well)


Reviewer 6

Summary

This paper deals with reconstructing the graph of an Ising model. The paper is well-written. The main effort is to improve upon the recent results provided by Bresler, showing that the complexity of identifying the structure of max degree d Ising model is polynomial in p and independent of d. Strong Points: 1) The timeliness of the topic in this paper is good, meaning that there is currently ongoing interest and work on Ising model reconstruction. 2) The paper contains more or less all the necessary theoretical results (mainly in terms of sample complexity) that one would expect to see in such a draft. 3) The approach to the solution, i.e., the use of the ISO and the l_1-penalization is quite intuitive in the way that is described in the paper. Weak points: 1) The whole approach is based on the introduction of the ISO. This is the main trick in the proposed approach. Other than that, the rest of the method and its analysis are usual and well studied (l_1-penalization and connection with the tutorial by Negahban et. al. on M-estimators), i.e., the rest of the proposed approach is nothing but very usual and well-studied approaches and ideas. 2) Due to the previous point, the analysis is based on well-established proof techniques or even more crucially similar mathematical statements that have been extensively derived and proposed in the related literature. 3) The authors need to highlight better the discussion in lines 166-170. 4) I would like to see more simulations demonstrating the performance of the proposed method. Overall, I think that the paper is nice, nevertheless with the aforementioned weaknesses under consideration.

Qualitative Assessment

The timeliness of the topic and the derived results are more important according to my opinion than the aforementioned weaknesses. Although normally I would not suggest publication of papers with l_1-penalization, since I think that these approaches have been extensively studied so far in literature, in this particular case I think that an exemption can be made.

Confidence in this Review

2-Confident (read it all; understood it all reasonably well)